# Neural Differential Equations for Learning to Program Neural Nets Through Continuous Learning Rules

**Kazuki Irie**[1]  **Francesco Faccio**[1]  **Jürgen Schmidhuber**[1,2]
[1]The Swiss AI Lab, IDSIA, USI & SUPSI, Lugano, Switzerland
[2]AI Initiative, KAUST, Thuwal, Saudi Arabia
`{kazuki, francesco, juergen}@idsia.ch`

## Abstract

Neural ordinary differential equations (ODEs) have attracted much attention as continuous-time counterparts of deep residual neural networks (NNs), and numerous extensions for recurrent NNs have been proposed. Since the 1980s, ODEs have also been used to derive theoretical results for NN learning rules, e.g., the famous connection between Oja's rule and principal component analysis. Such rules are typically expressed as additive iterative update processes which have straightforward ODE counterparts. Here we introduce a novel combination of learning rules and Neural ODEs to build continuous-time sequence processing nets that learn to manipulate short-term memory in rapidly changing synaptic connections of other nets. This yields continuous-time counterparts of Fast Weight Programmers and linear Transformers. Our novel models outperform the best existing Neural Controlled Differential Equation based models on various time series classification tasks, while also addressing their fundamental scalability limitations. Our code is public.[1]

## 1  Introduction

Neural ordinary differential equations (NODEs) [1] have opened a new perspective on continuous-time computation with neural networks (NNs) as a practical framework for machine learning based on differential equations. While the original approach—proposed as a continuous-depth version of deep feed-forward residual NNs [2, 3]—only covers autonomous ODEs entirely determined by the initial conditions, more recent extensions deal with sequential data (reviewed in Sec. 2.1) in a way similar to what is typically done with standard recurrent NNs (RNNs) in the discrete-time scenario. This potential for continuous-time (CT) sequence processing (CTSP) is particularly interesting, since there are many applications where datapoints are observed at irregularly spaced time steps, and CT sequence models might better deal with such data than their discrete-time counterparts. However, the development of NODEs for CTSP is still at an early stage. For example, a popular approach of Neural Controlled Differential Equations [4] (NCDEs; also reviewed in Sec. 2.1) has in practice only one architectural variant corresponding to the "vanilla" RNN [5]. Discrete-time processing, however, exploits many different RNN architectures as well as Transformers [6].

While it is not straightforward to transform the standard Transformer into a CT sequence processor, we'll show that the closely related Fast Weight Programmers (FWPs) [7, 8, 9, 10] and linear Transformers [11] (reviewed in Sec. 2.3) have direct CT counterparts. In FWPs, temporal processing of short-term memory (stored in fast weight matrices) uses learnable sequences of *learning rules*. Hence CT versions of FWPs will require differential equations to model the learning rules. This relates to a trend of the 1980s/90s. Among many old connections between NNs and dynamical systems described by ODEs (e.g., [12, 13, 14, 15, 16, 17]), the theoretical analysis of NN learning rules in the ODE

---

36th Conference on Neural Information Processing Systems (NeurIPS 2022).

framework has been particularly fruitful. Consider the famous example of Oja's rule [18] (briefly reviewed in Sec. 2.2): many results on its stability, convergence, and connection to Principal Component Analysis [19, 20] were obtained using its ODE counterpart (e.g., [18, 21, 22, 23, 24, 25, 26, 27]).

Here we propose a novel combination of Neural ODEs and learning rules, to obtain a new class of sequence processing Neural ODEs which are continuous-time counterparts of Fast Weight Programmers and linear Transformers. The resulting models are general-purpose CT sequence-processing NNs, which can directly replace the standard Neural CDE models typically used for supervised CT sequence processing tasks. To the best of our knowledge, there is no previous work on Neural ODE-based Transformer families for CT sequence processing, despite their popularity in important types of discrete time computation such as Natural Language Processing and beyond. We also show how our approach solves the fundamental limitation of existing Neural CDEs in terms of model size scalability.

We conduct experiments on three standard time series classification tasks covering various scenarios (regularly sampled, irregularly sampled with missing values, and very long time series). We demonstrate that our novel models outperform existing Neural ODE-based sequence processors, in some cases by a large margin.

## 2 Background

We briefly review the main background concepts this work builds upon: NODEs for sequence processing (Sec. 2.1), NN learning rules and their connection to ODEs (Sec. 2.2), and Fast Weight Programmers whose memory update is based on learning rules controlled by an NN (Sec. 2.3).

### 2.1 Neural ODEs (NODEs) and Their Extensions for Sequence Processing

Here we review the core idea of NODEs [1]. In what follows, let $n$, $N$, $d$, $d_{\text{in}}$ denote positive integers, $T$ be a positive real number, and $\theta$ denote an arbitrary set of real numbers. We consider a residual layer (say, the $n$-th layer with a dimension $d$) in an $N$-layer deep NN which transforms an input $\boldsymbol{h}_{n-1} \in \mathbb{R}^d$ to an output $\boldsymbol{h}_n \in \mathbb{R}^d$ with a parameterised function $\boldsymbol{f}_\theta : \mathbb{R}^d \to \mathbb{R}^d$ as follows:

$$\boldsymbol{h}_n = \boldsymbol{h}_{n-1} + \boldsymbol{f}_\theta(\boldsymbol{h}_{n-1}) \tag{1}$$

This coincides [28, 29, 30, 31, 32, 33, 34, 1] with the following equation for $\epsilon = 1$

$$\boldsymbol{h}(t_n) = \boldsymbol{h}(t_{n-1}) + \epsilon \boldsymbol{f}_\theta(\boldsymbol{h}(t_{n-1})) \tag{2}$$

where $\boldsymbol{h} : [0, T] \to \mathbb{R}^d$ is a function such that $\boldsymbol{h}(t_n) = \boldsymbol{h}_n$ holds for all $n : 0 \leq n \leq N$ and $t_n \in [0, T]$ such that $t_n - t_{n-1} = \epsilon > 0$ if $n \geq 1$. This equation is a forward Euler discretisation of the ordinary differential equation defined for all $t \in (t_0, T]$ as

$$\boldsymbol{h}'(t) = \boldsymbol{f}_\theta(\boldsymbol{h}(t)) \quad \text{or} \quad \boldsymbol{h}(t) = \boldsymbol{h}(t_0) + \int_{s=t_0}^{t} \boldsymbol{f}_\theta(\boldsymbol{h}(s)) ds \tag{3}$$

where $\boldsymbol{h}'$ denotes the first order derivative. This establishes the connection between the ODE and the deep residual net with parameters $\theta$ shared across layers[2]: given the initial condition $\boldsymbol{h}(t_0) = \boldsymbol{h}_0$, the solution to this equation evaluated at time $T$, i.e., $\boldsymbol{h}(T)$, corresponds to the output of this deep residual NN, which can be computed by an ODE solver. We denote it as a function ODESolve taking four variables: $\boldsymbol{h}(T) = \text{ODESolve}(\boldsymbol{f}_\theta, \boldsymbol{h}_0, t_0, T)$. During training, instead of backpropagating through the ODE solver's operations, the continuous *adjoint sensitivity method* [35] (which essentially solves another ODE but backward in time) can compute gradients with $O(d)$ memory requirement, constant w.r.t. $T$ [1].

A natural next step is to extend this formulation for RNNs, i.e., the index $n$ now denotes the time step, and we assume an external input $\boldsymbol{x}_n \in \mathbb{R}^{d_{\text{in}}}$ at each step $n$ to update the hidden state $\boldsymbol{h}_{n-1}$ to $\boldsymbol{h}_n$ as

$$\boldsymbol{h}_n = \boldsymbol{f}_\theta(\boldsymbol{h}_{n-1}, \boldsymbol{x}_n) \tag{4}$$

Depending on the property of external inputs $(\boldsymbol{x}_n)_{n=1}^N = (\boldsymbol{x}_1, ..., \boldsymbol{x}_N)$, there are different ways of defining NODEs for sequence processing. We mainly distinguish three cases.

---

[2]Or we make $\theta$ dependent of $t$ such that parameters are "depth/layer-dependent" as in standard deep nets.

**First**, when there is a possibility to construct a *differentiable* control signal $\boldsymbol{x} : t \mapsto \boldsymbol{x}(t) \in \mathbb{R}^{d_{\text{in}}}$ for $t \in [t_0, T]$ from the inputs $(\boldsymbol{x}_n)_{n=1}^N$; an attractive approach by Kidger et al. [4] handles the corresponding dynamics in a *neural controlled differential equation* (NCDE):

$$\boldsymbol{h}(t) = \boldsymbol{h}(t_0) + \int_{s=t_0}^t \boldsymbol{F}_\theta(\boldsymbol{h}(s))d\boldsymbol{x}(s) = \boldsymbol{h}(t_0) + \int_{s=t_0}^t \boldsymbol{F}_\theta(\boldsymbol{h}(s))\boldsymbol{x}'(s)ds \tag{5}$$

where $\boldsymbol{F}_\theta$ is a parameterised function (typically a few-layer NN) which maps a vector $\boldsymbol{h}(s) \in \mathbb{R}^d$ to a matrix $\boldsymbol{F}_\theta(\boldsymbol{h}(s)) \in \mathbb{R}^{d \times d_{\text{in}}}$ (we already relate this component to Recurrent Fast Weight Programmers below) and thus, $\boldsymbol{F}_\theta(\boldsymbol{h}(s))d\boldsymbol{x}(s)$ denotes a matrix-vector multiplication. There are several methods to construct the *control* $\boldsymbol{x} : [t_0, T] \to \mathbb{R}^{d_{\text{in}}}$ based on the discrete data points $(\boldsymbol{x}_n)_{n=1}^N$, such that its differentiability is guaranteed. In this work, we follow Kidger et al. [4] and mainly use natural cubic splines over all data points (which, however, makes it incompatible with auto-regressive processing); for better alternatives, we refer to Morrill et al. [36]. Since the final equation is again an NODE with a vector field of form $g_{\theta,\boldsymbol{x}'}(s, \boldsymbol{h}(s)) = \boldsymbol{F}_\theta(\boldsymbol{h}(s))\boldsymbol{x}'(s)$, all methods described above are applicable: ODE solver for evaluation and continuous adjoint method for memory efficient training. A notable extension of Neural CDEs is the use of *log-signatures* to sub-sample the input sequence [37]. The resulting NCDEs are called *neural rough differential equations* (NRDEs), which are relevant for processing long sequences. One fundamental limitation of the NCDEs above is the lack of scalability of $\boldsymbol{F}_\theta : \mathbb{R}^d \to \mathbb{R}^{d \times d_{\text{in}}}$. For example, if we naively parameterise $\boldsymbol{F}_\theta$ using a linear layer, the size of its weight matrix is $d^2 * d_{\text{in}}$ which quadratically increases with the hidden state size $d$. Previous attempts [38] do not successfully resolve this issue without performance degradation. In Sec. 5, we'll discuss how our models (Sec. 3.2) naturally circumvent this limitation while remaining powerful NCDEs.

On a side note, the NCDE is often referred to as the "continuous-time analogue" to RNNs [4], but this is a bit misleading: discrete-time RNN equations corresponding to the continuous-time Eq. 5 do not reflect the standard RNN of Eq. 4 but:

$$\boldsymbol{h}_n = \boldsymbol{h}_{n-1} + \boldsymbol{W}_{n-1}(\boldsymbol{x}_n - \boldsymbol{x}_{n-1}) \tag{6}$$
$$\boldsymbol{W}_n = \boldsymbol{F}_\theta(\boldsymbol{h}_n) \tag{7}$$

where one network (Eq. 6) learns to translate the variation of inputs $(\boldsymbol{x}_n - \boldsymbol{x}_{n-1})$ into a change in the state space, using a weight matrix $\boldsymbol{W}_{n-1}$ which itself is generated by another network ($\boldsymbol{F}_\theta : \mathbb{R}^d \to \mathbb{R}^{d \times d_{\text{in}}}$; Eq. 7) on the fly from the hidden state. This model is thus a kind of Recurrent FWP [39, 40, 10].

**Second**, even if $\boldsymbol{x}$ is not differentiable, having access to (piece-wise) continuous $\boldsymbol{x}$ defined and bounded over an interval of interest $[t_0, T]$ is enough to define a sequence processing NODE, by making it part of the vector field:

$$\boldsymbol{h}(t) = \boldsymbol{h}(t_0) + \int_{s=t_0}^t \boldsymbol{f}_\theta(\boldsymbol{h}(s), \boldsymbol{x}(s))ds \tag{8}$$

where the vector field $\boldsymbol{f}_\theta(\boldsymbol{h}(t), \boldsymbol{x}(t)) = g_{\theta,\boldsymbol{x}}(t, \boldsymbol{h}(t))$ can effectively be evaluated at any time $t \in [t_0, T]$. We refer to this second approach as a *direct NODE* method. While Kidger et al. [4] theoretically and empirically show that this approach is less expressive than the NCDEs above, we'll show how in our case of learning rules one can derive interesting models within this framework, which empirically perform on par with the CDE variants.

**Finally**, when no control function with one of the above properties can be constructed, a mainstream approach dissociates the continuous-time hidden state update via ODE for the time between two observations (e.g., Eq. 9 below) from integration of the new data (Eq. 10 below). Notable examples of this category include ODE-RNNs [41, 42] which transform the hidden states $\boldsymbol{h}_{n-1}$ to $\boldsymbol{h}_n$ for each observation $\boldsymbol{x}_n$ available at time $t_n$ as follows:

$$\boldsymbol{u}_n = \text{ODESolve}(\boldsymbol{f}_{\theta_1}, \boldsymbol{h}_{n-1}, t_{n-1}, t_n) \tag{9}$$
$$\boldsymbol{h}_n = \boldsymbol{\phi}_{\theta_2}(\boldsymbol{x}_n, \boldsymbol{u}_n) \tag{10}$$

where Eq. 9 autonomously updates the hidden state between two observations using a function $\boldsymbol{f}_{\theta_1}$ parameterised by $\theta_1$, while in Eq. 10, function $\boldsymbol{\phi}_{\theta_2}$ parameterised by $\theta_2$ integrates the new input $\boldsymbol{x}_n$ into the hidden state. In Latent ODE-RNN [41], a popular extension of this approach to the variational setting, the initial recurrent state $\boldsymbol{h}_0$ is sampled from a prior (during training, an additional encoder is trained to map sequences of inputs to parameters of the prior). While this third case is not our focus, we'll also show how to use FWPs in this scenario in Sec. 3.3 for the sake of completeness.

## 2.2 Learning Rules and Their Connections to ODEs

Learning rules of artificial NNs describe the process which modifies their weights in response to some inputs. This includes the standard backpropagation rule (also known as the reverse mode of automatic differentiation) derived for the case of supervised learning, as well as rules inspired by Hebb's informal rule [43] in "unsupervised" settings. Here we focus on the latter. Let $n$, $d_{\text{in}}$ $d_{\text{out}}$ be positive integers. Given a linear layer with a weight matrix $\boldsymbol{W}_n \in \mathbb{R}^{d_{\text{out}} \times d_{\text{in}}}$ (the single output neuron case $d_{\text{out}} = 1$ is the focus of the classic works) at time $n$ which transforms input $\boldsymbol{x}_n \in \mathbb{R}^{d_{\text{in}}}$ to output $\boldsymbol{y}_n \in \mathbb{R}^{d_{\text{out}}}$ as

$$\boldsymbol{y}_n = \boldsymbol{W}_{n-1}\boldsymbol{x}_n \tag{11}$$

the pure Hebb-style additive learning rule modifies the weights according to

$$\boldsymbol{W}_n = \boldsymbol{W}_{n-1} + \eta_n \boldsymbol{y}_n \otimes \boldsymbol{x}_n \tag{12}$$

where $\otimes$ denotes outer product and $\eta_n \in \mathbb{R}_+$ is a learning rate at time $n$.

Oja [18] proposed stability improvements to this rule through a decay term

$$\boldsymbol{W}_n = \boldsymbol{W}_{n-1} + \eta_n \boldsymbol{y}_n \otimes (\boldsymbol{x}_n - \boldsymbol{W}_{n-1}^\top \boldsymbol{y}_n) \tag{13}$$

whose theoretical analysis has since the 1980s been a subject of many researchers covering stability, convergence, and relation to Principal Component Analysis [18, 21, 22, 23, 24, 25, 26, 27, 44]. One key approach for such theoretical analysis is to view the equation above as a discretisation of the following ODE:

$$\boldsymbol{W}'(t) = \eta(t)\boldsymbol{y}(t) \otimes (\boldsymbol{x}(t) - \boldsymbol{W}(t-1)^\top \boldsymbol{y}(t)) \tag{14}$$

On a related note, studies of RNNs (e.g., [45, 46]) or learning dynamics (e.g., [47]) have also profited from ODEs.

## 2.3 Fast Weight Programmers & Linear Transformers

Fast Weight Programmers (FWP; [7, 8, 9, 10]) are general-purpose (auto-regressive) sequence processing NNs. In general, an FWP is a system of two NNs: a *slow* NN, the *programmer*, rapidly generates during runtime weight changes of another neural network, the *fast* NN. The (slow) weights of the slow net are typically trained by gradient descent. Variants of FWPs whose weight generation is based on outer products between keys and values [7] have been shown [9] to be equivalent to Linear Transformers [11] (using the mathematical equivalence known from perceptron/kernel machine duality [48, 49]). These FWPs use sequences of learning rules to update short-term memory in form of a fast weight matrix. A practical example of such FWPs is the DeltaNet [9] which transforms an input $\boldsymbol{x}_n \in \mathbb{R}^{d_{\text{in}}}$ into an output $\boldsymbol{y}_n \in \mathbb{R}^{d_{\text{out}}}$ at each time step $n$ while updating its fast weight matrix $\boldsymbol{W}_{n-1} \in \mathbb{R}^{d_{\text{out}} \times d_{\text{key}}}$ as follows:

$$\beta_n, \boldsymbol{q}_n, \boldsymbol{k}_n, \boldsymbol{v}_n = \boldsymbol{W}_{\text{slow}}\boldsymbol{x}_n \tag{15}$$

$$\boldsymbol{W}_n = \boldsymbol{W}_{n-1} + \sigma(\beta_n)(\boldsymbol{v}_n - \boldsymbol{W}_{n-1}\phi(\boldsymbol{k}_n)) \otimes \phi(\boldsymbol{k}_n) \tag{16}$$

$$\boldsymbol{y}_n = \boldsymbol{W}_n \phi(\boldsymbol{q}_n) \tag{17}$$

where the slow net (Eq. 15; with weights $\boldsymbol{W}_{\text{slow}} \in \mathbb{R}^{(1+2*d_{\text{key}}+d_{\text{out}}) \times d_{\text{in}}}$) generates key/value vectors $\boldsymbol{k}_n \in \mathbb{R}^{d_{\text{key}}}$ and $\boldsymbol{v}_n \in \mathbb{R}^{d_{\text{out}}}$ as well as a scalar $\beta_n \in \mathbb{R}$ to obtain a dynamic learning rate by applying a sigmoid function $\sigma$, and $\phi$ is an element-wise activation function whose output elements are positive and sum up to one (typically softmax). These fast dynamic variables generated by a slow NN are used in a learning rule (Eq. 16) akin to the classic delta rule [50] to update the fast weight matrix. The output is finally produced by the forward computation of the fast NN, i.e., by *querying* the fast weight matrix by the generated query vector $\boldsymbol{q}_n \in \mathbb{R}^{d_{\text{key}}}$ (Eq. 17). An intuitive interpretation of the fast weight matrix is a key-value associative memory with write and read operations defined by Eq. 16 and 17, respectively. This encourages intuitive thoughts about memory capacity (limited by the number of "keys" we can store without interference) [9]. For instance, if we replace the learning rule (i.e., memory writing operation) of Eq. 16 by a pure additive Hebb-style rule (and a fixed learning rate of 1.0): $\boldsymbol{W}_n = \boldsymbol{W}_{n-1} + \boldsymbol{v}_n \otimes \phi(\boldsymbol{k}_n)$, we obtain the Linear Transformer [11] (we refer to prior work [9] for further explanations of the omission of attention normalisation). Such a purely additive learning rule often suffers from long term dependencies, unlike the delta rule [9]. We'll confirm this trend also in the CT models (using the EigenWorms dataset). For later convenience, we introduce a notation FWP which denotes generic FWP operations: $\boldsymbol{y}_n, \boldsymbol{W}_n = \text{FWP}(\boldsymbol{x}_n, \boldsymbol{W}_{n-1}; \boldsymbol{W}_{\text{slow}})$.

# 3 Continuous-Time Fast Weight Programmers

We propose continuous-time counterparts of Fast Weight Programmers (Sec. 2.3) which naturally combine ODEs for learning rules (Sec. 2.2) and existing approaches for sequence processing with NODEs (Sec. 2.1). We present three types of these CT FWP models in line with the categorisation of Sec. 2.1 while the main focus of this work is on the two first cases.

## 3.1 Direct NODE-based FWPs

In the *direct NODE* approach (reviewed in Sec. 2.1), we assume a (piece-wise) continuous control signal $\boldsymbol{x} : t \mapsto \boldsymbol{x}(t)$ bounded over an interval $[t_0, T]$. We make it part of the vector field to define an ODE describing a *continuous-time learning rule* for a fast weight matrix $\boldsymbol{W}(t)$:

$$\boldsymbol{W}(t) = \boldsymbol{W}(t_0) + \int_{s=t_0}^{t} \boldsymbol{F}_\theta(\boldsymbol{W}(s), \boldsymbol{x}(s)) ds \tag{18}$$

where $\boldsymbol{W} : t \mapsto \boldsymbol{W}(t) \in \mathbb{R}^{d_{\text{out}} \times d_{\text{key}}}$ is a function defined on $[t_0, T]$, and $\boldsymbol{F}_\theta$ is an NN parameterised by $\theta$ which maps onto $\mathbb{R}^{d_{\text{out}} \times d_{\text{key}}}$. This is a neural differential equation for learning to program a neural net through continuous learning rules, that is, to train a fast weight matrix $\boldsymbol{W}(t)$ of a fast NN (Eq. 20 below) for each sequential control $\boldsymbol{x}$. Like in the discrete-time FWPs (Sec. 2.3), the output $\boldsymbol{y}(T) \in \mathbb{R}^{d_{\text{out}}}$ is obtained by *querying* this fast weight matrix[3] (e.g., at the last time step $T$):

$$\boldsymbol{q}(T) = \boldsymbol{W}_q \boldsymbol{x}(T) \tag{19}$$
$$\boldsymbol{y}(T) = \boldsymbol{W}(T) \boldsymbol{q}(T) \tag{20}$$

where $\boldsymbol{W}_q \in \mathbb{R}^{d_{\text{key}} \times d_{\text{in}}}$ is a slow weight matrix used to generate the query $\boldsymbol{q}(T) \in \mathbb{R}^{d_{\text{key}}}$ (Eq. 19). Now we need to specify $\boldsymbol{F}_\theta$ in Eq. 18 to fully define the learning rule. We focus on three variants:

$$\boldsymbol{F}_\theta(\boldsymbol{W}(s), \boldsymbol{x}(s)) = \sigma(\beta(s)) \begin{cases} \boldsymbol{k}(s) \otimes \boldsymbol{v}(s) & \text{Hebb-style} \\ \boldsymbol{v}(s) \otimes \big(\boldsymbol{k}(s) - \boldsymbol{W}(s)^\top \boldsymbol{v}(s)\big) & \text{Oja-style} \\ \big(\boldsymbol{v}(s) - \boldsymbol{W}(s)\boldsymbol{k}(s)\big) \otimes \boldsymbol{k}(s) & \text{Delta-style} \end{cases} \tag{21}$$

where $[\beta(s), \boldsymbol{k}(s), \boldsymbol{v}(s)] = \boldsymbol{W}_{\text{slow}} \boldsymbol{x}(s)$ with a slow weight matrix $\boldsymbol{W}_{\text{slow}} \in \mathbb{R}^{(1+d_{\text{key}}+d_{\text{out}}) \times d_{\text{in}}}$. As in the discrete-time FWP (Sec. 2.3), the slow NN generates $\beta(s) \in \mathbb{R}$ (to which we apply the sigmoid function $\sigma$ to obtain a learning rate), key $\boldsymbol{k}(s) \in \mathbb{R}^{d_{\text{key}}}$ and value $\boldsymbol{v}(s) \in \mathbb{R}^{d_{\text{out}}}$ vectors from input $\boldsymbol{x}(s)$. These variants are inspired by the respective classic learning rules of the same name, while they are crucially different from the classic ones in the sense that all variables involved (key, value, learning rate) are continually generated by the slow NN. In the experimental section, we'll comment on how some of these design choices can result in task-dependent performance gaps. In practice, we use the *multi-head* version of the operations above (i.e., by letting $H$ be a positive integer denoting the number of heads, query/key/value vectors are split into $H$ sub-vectors and Eqs. 20-21 are conducted independently for each head). The output is followed by the standard feed-forward block like in Transformers [6]. Possible extensions for deeper models are discussed in Appendix C.1.

## 3.2 NCDE-based FWPs

Here we present models based on NCDEs (reviewed in Sec. 2.1). We assume availability of a differentiable control signal $\boldsymbol{x}(t)$, whose first order derivative is denoted by $\boldsymbol{x}'(t)$. Given the NCDE formulation of Eq. 5, the most straight-forward approach to obtain a CT Fast Weight Programmer is to extend the dimensionality of the recurrent hidden state, i.e., we introduce a parameterised function $\mathsf{F}_\theta$ which maps a matrix $\boldsymbol{W}(t) \in \mathbb{R}^{d_{\text{out}} \times d_{\text{key}}}$ to a third-order tensor $\mathsf{F}_\theta(\boldsymbol{W}(t)) \in \mathbb{R}^{d_{\text{out}} \times d_{\text{key}} \times d_{\text{in}}}$:

$$\boldsymbol{W}(t) = \boldsymbol{W}(t_0) + \int_{s=t_0}^{t} \mathsf{F}_\theta(\boldsymbol{W}(s)) d\boldsymbol{x}(s) = \boldsymbol{W}(t_0) + \int_{s=t_0}^{t} \mathsf{F}_\theta(\boldsymbol{W}(s)) \boldsymbol{x}'(s) ds \tag{22}$$

However, this approach is obviously not scalable since the input and output dimensions ($d_{\text{out}} \times d_{\text{key}}$ and $d_{\text{out}} \times d_{\text{key}} \times d_{\text{in}}$) of $\mathsf{F}_\theta$ can be too large in practice. A more tractable CDE-based approach can

---

[3]In practice, we also apply element-wise activation functions to query/key/value vectors where appropriate, which we omit here for readability. We refer to Appendix A for further details.

be obtained by providing $\boldsymbol{x}$ and/or $\boldsymbol{x}'$ to the vector field:

$$\boldsymbol{W}(t) = \boldsymbol{W}(t_0) + \int_{s=t_0}^{t} \mathsf{F}_\theta(\boldsymbol{W}(s), \boldsymbol{x}(s), \boldsymbol{x}'(s))\boldsymbol{x}'(s)ds \tag{23}$$

While this equation still remains a CDE because of the multiplication from the right by $d\boldsymbol{x} = \boldsymbol{x}'(s)ds$, the additional inputs to the vector field offer a way of making use of various learning rules, as in the case of direct NODE approach above (Sec. 3.1). To be specific, either $\boldsymbol{x}$ and $\boldsymbol{x}'$ or only $\boldsymbol{x}'$ is required in the vector field to obtain these tractable CDEs. Here we present the version which uses both $\boldsymbol{x}$ and $\boldsymbol{x}'$ [4]. The resulting vector fields for different cases are:

$$\mathsf{F}_\theta\big(\boldsymbol{W}(s), \boldsymbol{x}(s), \boldsymbol{x}'(s)\big)\boldsymbol{x}'(s) = \sigma(\beta(s)) \begin{cases} \boldsymbol{W}_k\boldsymbol{x}(s) \otimes \boldsymbol{W}_v\boldsymbol{x}'(s) & \text{Hebb} \\ \big(\boldsymbol{W}_k\boldsymbol{x}(s) - \boldsymbol{W}(s)^\top \boldsymbol{W}_v\boldsymbol{x}'(s)\big) \otimes \boldsymbol{W}_v\boldsymbol{x}'(s) & \text{Oja} \\ \big(\boldsymbol{W}_v\boldsymbol{x}(s) - \boldsymbol{W}(s)\boldsymbol{W}_k\boldsymbol{x}'(s)\big) \otimes \boldsymbol{W}_k\boldsymbol{x}'(s) & \text{Delta} \end{cases} \tag{24}$$

As can be seen above, the use of CDEs to describe a continuous fast weight learning rule thus naturally results in a key/value memory where $\boldsymbol{x}'$ is used to generate either key or value vectors. Because of the multiplication from the right by $\boldsymbol{x}'$, the role of $\boldsymbol{x}'$ changes depending on the choice of learning rule: $\boldsymbol{x}'$ is used to generate the key in the Delta case but the value vector in the case of Oja. In the case of Hebb, the choice made in Eq. 24 of using $\boldsymbol{x}$ for keys and $\boldsymbol{x}'$ for values is arbitrary since Eq. 24 is symmetric in terms of roles of keys and values (see an ablation study in Appendix C.2 for the other case where we use $\boldsymbol{x}'$ to generate the key and $\boldsymbol{x}$ for the value). The querying operation (analogous to Eqs. 19-20 for the direct NODE case) is also modified accordingly, depending on the choice of learning rule, such that the same input ($\boldsymbol{x}$ or $\boldsymbol{x}'$) is used to generate both key and query:

$$\boldsymbol{y}(T) = \begin{cases} \boldsymbol{W}(T)^\top \boldsymbol{W}_q\boldsymbol{x}(T) & \text{Hebb and Oja} \\ \boldsymbol{W}(T)\boldsymbol{W}_q\boldsymbol{x}'(T) & \text{Delta} \end{cases} \tag{25}$$

Note that since the proposed vector field $\mathsf{F}_\theta\big(\boldsymbol{W}(s), \boldsymbol{x}(s), \boldsymbol{x}'(s)\big)\boldsymbol{x}'(s)$ is more general than the one used in the original NCDE $\mathsf{F}_\theta\big(\boldsymbol{W}(s)\big)\boldsymbol{x}'(s)$, any theoretical results on the CDE remain valid (which, however, does not tell us anything about the best choice for its exact parameterisation).

## 3.3 ODE-RFWP and Latent ODE-RFWP

The main focus of this work is the setting of Kidger et al. [4] where we assume the existence of some control signal $\boldsymbol{x}$ (Sec. 3.1 and 3.2 above). However, here we also show a way of using FWPs in the third/last case presented in Sec. 2.1 where no control $\boldsymbol{x}(t)$ is available (or can be constructed), i.e., we only have access to discrete observations $(\boldsymbol{x}_n)_{n=0}^N$. Here we cannot directly define the vector field involving continuous transformations using the inputs. We follow the existing approaches (ODE-RNN or Latent ODE; Sec. 2.1) which use two separate update functions: A discrete recurrent state update is executed every time a new observation is available to the model, while a continuous update using an autonomous ODE is conducted in between observations. Unlike with standard recurrent state vectors, however, it is not practical to autonomously evolve high-dimensional fast weight matrices[5]. We therefore opt for using a Recurrent FWP (RFWP) [10] and combine it with an ODE:

$$\boldsymbol{u}_n = \text{ODESolve}(\boldsymbol{f}_{\theta_1}, \boldsymbol{h}_{n-1}, t_{n-1}, t_n) \tag{26}$$
$$\boldsymbol{h}_n, \boldsymbol{W}_n = \text{FWP}([\boldsymbol{x}_n, \boldsymbol{u}_n], \boldsymbol{W}_{n-1}; \theta_2) \tag{27}$$

where we keep the fast weight learning rule itself discrete (Eq. 27), but evolve the recurrent state vector $\boldsymbol{u}_n$ using an ODE (Eq. 26) such that the information to be read/written to the fast weight matrix is controlled by a variable which is continuously updated between observations. We refer to this model as ODE-RFWP and its variational variant as Latent ODE-RFWP.

Since we focus on continuous-time learning rules, the case above is not of central interest as the learning rule remains discrete here (Eq. 27). Nevertheless, in Appendix C.3, we also provide some experimental results for model-based reinforcement learning settings corresponding to this case.

---

[4] The equations for the version using only $\boldsymbol{x}'$ can be obtained by replacing $\boldsymbol{x}$ by $\boldsymbol{x}'$ in Eq. 24. We provide an ablation in Appendix C.2. As a side note, we also obtain the equation for the CDE using only $\boldsymbol{x}'$ by replacing $\boldsymbol{x}$ by $\boldsymbol{x}'$ in Eq. 18 for the direct NODE case.

[5] Such an approach would require a computationally expensive matrix-to-matrix transforming NN, whose scalability is limited.

Table 1: **Accuracy (%) on the Speech Commands classification task** and **AUC ($\times 10^2$) on the PhysioNet Sepsis prediction task**. PhysioNet has two cases: with (OI) or without (no-OI) observational intensity (see text for details). Numbers marked by * are taken from Kidger et al. [4]. Mean and standard deviation (std) are computed over 5 runs.

| Type | Model | Speech Commands | PhysioNet Sepsis | |
| --- | --- | --- | --- | --- |
| | | | OI | no-OI |
| Direct NODE | GRU-ODE [4]* | 47.9 (2.9) | 85.2 (1.0) | 77.1 (2.4) |
| | Hebb | 82.8 (1.1) | **90.4 (0.4)** | 82.9 (0.7) |
| | Oja | **85.4 (0.9)** | 88.9 (1.4) | 82.9 (0.5) |
| | Delta | 81.5 (3.8) | 89.8 (1.0) | **84.5 (2.9)** |
| CDE | NCDE [4]* | 89.8 (2.5) | 88.0 (0.6) | 77.6 (0.9) |
| | Hebb | 89.5 (0.3) | 89.9 (0.6) | **85.7 (0.3)** |
| | Oja | 90.0 (0.7) | **91.2 (0.4)** | 85.1 (2.5) |
| | Delta | **90.2 (0.2)** | 90.9 (0.2) | 84.5 (0.7) |

## 4 Experiments

We consider three datasets covering three types of time series which are regularly sampled (Speech Commands [51]), irregularly sampled with partially missing features (PhysioNet Sepsis [52]), or very long (EigenWorms [53]). We compare the proposed direct NODE and CDE based FWP models (Sec. 3.1 & 3.2) to NODE baselines previously reported on the same datasets [4, 36]. Appendix B provides further experimental details including hyper-parameters.

**Speech Commands.** The Speech Commands [51] is a single word speech recognition task. The datapoints are regularly sampled, and the sequence lengths are relatively short ($\leq 160$ frames), which makes this task a popular sanity check. Following prior work on NCDEs [4], we use 20 mel frequency cepstral coefficients as speech features and classify the resulting sequence to one out of ten keywords. The middle column of Table 1 shows the results. The table is split into the direct NODE (top) and CDE (bottom) based approaches. We first observe that among the direct NODE approaches, all our FWPs largely outperform ($\geq 80\%$ accuracy) the baseline GRU-ODE performance of 47.9% (the best direct NODE baseline from Kidger et al. [4]). This demonstrates that with a good parameterisation of the vector field, the direct NODE approach can achieve competitive performance. On the other hand, all CDE-based approaches yield similar performance. We also only see slight differences in terms of performance among different learning rules, without a clear winner for this task. This may indicate that the ordinary nature of this task (regularly sampled; short sequences) does not allow for differentiating among these CDE models, including the baseline.

**PhysioNet Sepsis.** The PhysioNet Sepsis is a dataset of the sepsis prediction task from the PhysioNet challenge 2019 [52]. This is again a dataset used by Kidger et al. [4] to evaluate NCDEs. The task is a binary prediction of sepsis from a time series consisting of measurements of 34 medical features (e.g., respiration rate) of patients' stays at an ICU. Each sequence is additionally labelled by five static features of the patient (e.g., age) which are fed to the model to generate the initial state of the ODE. Sequences are relatively short ($\leq 72$ frames) but datapoints are irregularly sampled and many entries are missing, which makes this task challenging. It comes in two versions: with and without the so-called *observation intensity* information (denoted as "OI" and "no-OI") which is one extra input feature indicating each observation's time stamp (providing the models with information on measurement frequency). This distinction is important since the prior work [4] has reported that existing ODE/CDE-based approaches struggle with the no-OI case of this task. Following the previous work, we report the performance in terms of Area Under the ROC Curve (AUC). The right part of Table 1 shows the results. We obtain large improvements in the no-IO case (from 77.6 to 85.7% for the CDEs and from 77.1 to 84.5% for the direct NODEs), while also obtaining small improvements in the OI case (from 85.2 to 90.4% for direct NODEs, and from 88.0 to 91.2% for CDEs). The no-OI performance of our models is also comparable to the best overall performance reported by Kidger [38]: 85.0 % (1.3) achieved by GRU-D [54]. This demonstrates the efficacy of

Table 2: **Classification Accuracy (%) on the EigenWorms task.** Numbers marked by * are taken from Morrill et al. [37]. Mean and standard deviation (std) are computed over 5 runs. "Sig-Depth" indicates the depth of the signature (with this number equal to 1, an RDE is reduced to a CDE). To facilitate comparisons to prior work [37], we also add the column "Step" indicating the sequence down-sampling factor (even if we fix it to the best value [37] of 4).

| Model | Sig-Depth | Step | Test Acc. [%] | |
|---|---|---|---|---|
| NRDE [37]* | 2 | 4 | 83.8 | (3.0) |
| Hebb | 2 | 4 | 45.6 | (5.9) |
| Oja | | | 46.7 | (7.5) |
| Delta | | | **87.7** | **(1.9)** |
| NCDE [37]* | 1 | 4 | 66.7 | (11.8) |
| Hebb | 1 | 4 | 41.0 | (6.5) |
| Oja | | | 49.7 | (9.9) |
| Delta | | | **91.8** | (3.4) |

CT FWP model variants for handling irregularly sampled data with partially missing features even in the case without frequency information. Differences between various learning rules are rather small again. In some cases, we observe performance to be very sensitive to hyper-parameters. For example, the best Oja-CDE configuration achieves 85.1% (2.5) with a learning rate of 6e-5, while this goes down to 79.6% (4.7) when the learning rate is changed to 5e-5.

**EigenWorms.** The EigenWorms dataset (which is part of the UEA benchmark [53]) is a 5-way classification of roundworm types based on time series tracking their movements. To be more specific, motions of a worm are represented by six features corresponding to their projections to six template movement shapes, called "eigenworms." While this dataset contains only 259 examples, it is notable for its very long sequences (raw sequence lengths exceed 17 K) and long-span temporal dependencies [37, 55, 56]. We use the same train/validation/test split ratio as the prior work [37] which reports Neural RDEs (NRDEs) as achieving the best NODE model performance on this dataset. The equations of our CT FWPs for the RDE case can be straightforwardly obtained by replacing the input $x$ in Eqs. 18-19 of the direct NODE formulation (or[6], $x$ and $x'$ in Eqs. 23-25 of the NCDEs) by the corresponding log-signatures. Table 2 shows the results, where "Step" denotes the time sub-sampling rate which is fixed to 4 for which the prior work [37] reports the best NRDE and NCDE performance. "Sig-Depth" denotes the depth of the log-signature (the deeper, the more log-signature terms we take into account, thus ending up with a larger input feature vector; we refer to the original paper [37] for further details). We consider two values for this parameter: 1 and 2. When set to 1, the input feature contains only the first derivative $x'(s)$ and thus the NRDE is reduced to an NCDE (with controls constructed via linear interpolation). We take the best NCDE performance from Morrill et al. [37] as the depth-1 baseline. Morrill et al. [37] report the best overall performance for the depth-2 NRDE (depth-2 baseline in our table). In both cases, we first note a large performance gap between models with different learning rules. While the naive Hebb and Oja based models struggle with this very long sequence processing (sequence length still exceeds 4 K with a down-sampling step size of 4), the Delta rule performs very well. This confirms the prior result in the discrete-time domain [9] which motivated the Delta rule design by its potential for handling long sequences (we refer to prior work [9] for further explanations). Since its performance on other tasks is comparable to the one of Hebb and Oja variants, the Delta rule is a natural default choice for parameterising the CT FWPs.

In both the depth-1 and depth-2 cases, we obtain large improvements compared to the respective baselines. It is counter-intuitive to find certain depth-2 models underperforming their depth-1 counter-parts, but this trend has also been observed in the original NRDEs [36]. Our best overall performance is obtained in the depth-1 case: 91.8 % (3.4) exceeds the previous best NRDE based model per-

---

[6]Conceptually these two approaches are equivalent: the direct NODE and NCDE coincide here. In practice, there can be a subtle difference due to an implementation detail. The direct NODE approach can apply layer normalisation to the input fed to *both* key and value projections (as they are both inside the vector field). In this case (as in our implementation), the corresponding NCDE formulation we obtain is based on the normalised input.

formance of 83.8 % (3.0) [36]. This almost matches the state-of-the-art accuracy of 92.8 % (1.8) reported by Rusch et al. [56] (using an ODE-inspired discrete-time model). Our model's performance variance is high (best single seed performance is 97.4% while the standard deviation is 3.4). The wall clock time is similar for our best model (last row in Table 2) and the NRDE baseline (36s/epoch on a GeForce RTX 2080 Ti) and their sizes are comparable (87 K vs. 65 K parameters respectively).

## 5    Discussions

**Scalability Advantage Compared to Standard NCDEs.**    In addition to the good empirical results shown above, our FWP approach also addresses an important limitation of existing NCDEs [4]: their scalability in terms of model size. The vector field in standard NCDEs (Eq. 5) requires an NN $F_\theta$ which takes a vector $h(s) \in \mathbb{R}^d$ as an input to produce a matrix of size $\mathbb{R}^{d \times d_{\text{in}}}$. This can be very challenging when $d_{\text{in}}$ or/and $d$ is large. Actually, the same bottleneck is present in the weight generation of FWPs [7]. The use of outer products can remediate this issue in discrete FWPs as well as in CT FWPs: the computations in our FWP-based NODE/NCDEs only involve "first-order" dimensions (i.e., no multiplication between different dimensions, such as $d \times d_{\text{in}}$) for NN outputs. This can scale well with increased model size, making feasible larger scale tasks infeasible for existing NCDEs. On the other hand, Kidger et al. [4] report that using outer products (in their "Sec. 6.1 on limitations") in standard NCDEs does not perform well. Why do outer products work well in our models but not in the original NCDEs? The answer may be simple. In the original NCDEs (Eq. 5), multiplications occur at each (infinitesimal) time step between the generated rank-one weight matrix $F_\theta(h(s))$ and $x'(s)$ before the sum. All these transformations are thus of rank one while we expect expressive transformations to be necessary to translate $x'(s)$ into changes in the state space. In contrast, in our CT FWPs, the ODE only parameterises the weight generation process of another net, and thus the rank-one matrices are never used in isolation: they are summed up over time (Eq. 18 or 23) to form an expressive weight matrix which is only then used for matrix multiplication (Eq. 20 or 25. The proposed FWP-NODE/NCDEs thus offer scalable alternatives to existing NCDEs, also yielding good empirical performance (Sec. 4).

**Importance of Memory Efficient Backpropagation for FWPs.**    Memory efficiency of continuous adjoint backpropagation may be not so important for standard NCDEs of state size $O(d)$, but is crucial for FWPs of state size $O(d^2)$ which can quickly become prohibitive for long sequences, as naive backpropagation stores all states used in the forward pass. Prior works on discrete FWPs [9, 57, 10] solve this problem by a custom memory-efficient implementation. Here, the continuous adjoint method naturally addresses this problem.

**Limitations.**    Our ablation studies and hyper-parameter tuning focus on optimising the model configuration/architecture. From the NODE perspective, other parameters may further improve performance or alleviate performance variability/stability issues observed in some cases (Sec. 4). For example, we use the numerical solver configurations of the baselines (see Appendix B) without tuning them. Similarly, we use natural cubic splines to construct the differentiable control signals for NCDEs in the Speech Commands and PhysioNet Sepsis tasks, following the original NCDE paper [4]. Morrill et al. [36] report performance enhancements by improving the corresponding interpolation methods (e.g., 93.7% on Speech Commands). Such further optimisation is not conducted here.

Generally speaking, real benefits of using continuous-time sequence processing models are yet to be proved. While we achieve improvements over the best existing NODE/NCDE models on multiple datasets, discrete-time models tailored to the corresponding problem still perform as well or even better than our improved CT models (e.g., Rusch et al. [56] for EigenWorms and Che et al. [54] for PhysioNet Sepsis; see Sec. 4).

**Related Work on Parameter/Weight ODEs.**    Other works use ODEs to parameterise the time-evolving weights of some model. However, they are limited to autonomous ODEs (i.e., no external control $x$ is involved). Zhang et al. [58] and Choromanski et al. [59] study coupled ODEs where one ODE is used for temporal evolution of parameters of the main Neural ODE. The scope of these two works is limited to autonomous ODEs corresponding to continuous-depth residual NNs with different parameters per depth. Deleu et al. [60] consider an ODE version of a gradient descent learning process for adaptation, but also formulated as an autonomous ODE. In contrast, our focus is really on

*sequence processing* where the model continuously receives external controls $x$ and translates them into weight changes of another network.

## 6  Conclusion

We introduced novel continuous-time sequence processing neural networks that learn to use sequences of ODE-based continuous learning rules as elementary programming instructions to manipulate short-term memory in rapidly changing synaptic connections of another network. The proposed models are continuous-time counterparts of Fast Weight Programmers and linear Transformers. Our new models experimentally outperform by a large margin existing Neural ODE based sequence processors on very long or irregularly sampled time series. Our Neural ODE/CDE based FWPs also address the fundamental scalability problem of the original Neural CDEs, which is highly promising for future applications of ODE based sequence processors to large scale problems.

## 7  Acknowledgements

We would like to thank Kidger et al. [4], Morrill et al. [37] and Du et al. [61] for their public code. This research was partially funded by ERC Advanced grant no: 742870, project AlgoRNN, and by Swiss National Science Foundation grant no: 200021_192356, project NEUSYM. We are thankful for hardware donations from NVIDIA and IBM. The resources used for this work were partially provided by Swiss National Supercomputing Centre (CSCS) project s1145 and s1154.

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
