# OpenReview forum: "Neural Differential Equations for Learning to Program Neural Nets Through Continuous Learning Rules"
_NeurIPS.cc/2022/Conference — NeurIPS 2022 Accept_

### Official Review · Reviewer_95a7 · 2022-07-08

**Rating:** 7
**Confidence:** 4
**Soundness:** 3 good
**Presentation:** 3 good
**Contribution:** 3 good

**Summary:**

This work introduces variants of Fast Weight Programmers, where the Fast Weight matrix W evolves via neural DEs, with vector field structures inspired by learning rules (namely Hebb, Oja, and Delta). These can be applied to continuous learning tasks.

The structure of the neural DEs here fall under a subclass considered by Kidger et al in the paper introducing neural CDEs (that is, the neural CDEs subsume these methods). However, this paper demonstrates empirically that the specific restriction to these models seem to perform well compared with optimising over the broader class of neural CDEs, as well as providing scalability benefits.

**Questions:**

In Section 3.3, it is assumed that the continuous control is not available and only discrete observations are accessible. In practice, this is almost always the case, in the paper by Morrill et al. [31], there is an extensive discussion on the construction of a continuous control path from the discrete observations. Perhaps the authors can highlight and clarify, when might one choose to take the discrete observations and apply methods in Section 3.3 rather than apply interpolation and use Neural CDEs?

**Limitations:**

The authors have discussed that some of their results have rather large variances, in particular Delta achieved the highest single seed performance for sepsis no-OI, but on average Hebb performed best in this case.

As discussed in weaknesses, the authors do not discuss comparisons with other baselines from non-ODE based neural networks. A discussion on where this work sits in the broader literature and any limitation of application of these new networks in broader problems would be beneficial.

Negative societal impact not applicable for this paper.

**Strengths And Weaknesses:**

This is a very nicely written paper, which presents the work done in a clear and concise manner. The background and literature is explained well. There is full transparency, the code is available for validation and reproduction of the results.

An observation is that this is pooling together a few ideas from different areas, so the improvement is more incremental, however this is not necessarily a weakness, and the empirical results and benefits seem to be good.

In terms of weaknesses, although it is not realistic to be comparing with all models out there, this paper does not discuss that in Kidger et al, the model that performed best for sepsis no-OI is in fact GRU-D, with a performance of 85% plus/minus 1.3, which is superior to the best performance of the three models introduced in this paper.

Furthermore, most of the comparison here is with Kidger et al. (2020), which has been superseded by Morrill et al. (2021) in Neural CDEs for online prediction tasks. It would be good for come comments and comparison with the results in the latter.

Some minor issues:
"ancient" for the 80s/90s can be slightly offensive!
Eqn (27) should be RFWP?
Table 1 should be AUC rather than AUR.

---

> ### Author Response · Authors · 2022-08-01
> **Response to Reviewer 95a7**
>
> We thank the reviewer for many positive comments.
>
> > *In terms of weaknesses, although it is not realistic to be comparing with all models out there, this paper does not discuss that in Kidger et al, the model that performed best for sepsis no-OI is in fact GRU-D, with a performance of 85\% plus/minus 1.3, which is superior to the best performance of the three models introduced in this paper.* //
> > *the authors do not discuss comparisons with other baselines from non-ODE based neural networks.*
>
> Thank you for pointing this out. This is a great point. In the revision, we will add a sentence to clearly state that all these models still underperform the discrete GRU-D model on the PhysioNet Sepsis/no IO case but also comment that our model largely reduces the gap between the two approaches.
> Please note, however, that for the EigenWorm experiments, we do already compare with the state-of-the-art results obtained by an ODE-inspired discrete-time RNN model (line 284 "This almost matches the state-of-the-art…"). So the comparison to non-ODE based methods is not completely missing in the submitted version (we opted for a concise comparison instead of a huge table, as a matter of style).
>
> > *Furthermore, most of the comparison here is with Kidger et al. (2020), which has been superseded by Morrill et al. (2021) in Neural CDEs for online prediction tasks. It would be good for come comments and comparison with the results in the latter.*
>
> Thank you for pointing this out. We will add the corresponding comments in the revision. Essentially our focus here is the base/off-line setting of Kidger et al. [4] (we also did not tune the interpolation strategy, which may lead to further improvements; as is also pointed out by Reviewer B6yz).
>
> > *Some minor issues: "ancient" for the 80s/90s can be slightly offensive!*
>
> We apologise for this. Reviewer B6yz made a similar comment. We just did not realise that the word "ancient" had such a negative connotation. We will simply remove it.
>
> > *Eqn (27) should be RFWP?*
>
> Here "FWP" is correct (as defined in line 150) because the "R" ("recurrent") part is covered by the dependency on $u_{n-1}$ which is a function of $h_{n-1}$.
>
> > *Table 1 should be AUC rather than AUR.*
>
> Thank you for pointing this out. We will fix this in the revision.
>
> > *In Section 3.3, it is assumed that the continuous control is not available and only discrete observations are accessible. In practice, this is almost always the case, in the paper by Morrill et al. [31], there is an extensive discussion on the construction of a continuous control path from the discrete observations. Perhaps the authors can highlight and clarify, when might one choose to take the discrete observations and apply methods in Section 3.3 rather than apply interpolation and use Neural CDEs?*
>
> By "continuous control not available", we refer to the case where it is not possible to construct it. The example case we have is the reinforcement learning setting for which we provide the experiments in Appendix C.3.
>
> We hope that our response resolves the reviewer's main concerns and provides clear answers to the questions.
> If that is the case, and if you think this work should be accepted, we'll appreciate it a lot if you can consider increasing the score. Thank you.

---

> > ### Author Response · Authors · 2022-08-09
> > **Friendly reminder**
> >
> > This is just a friendly reminder about the NeurIPS rebuttal deadline today. Thank you!

---

> > > ### Comment · Reviewer_95a7 · 2022-08-09
> > > **Response**
> > >
> > > Thank you to the author(s) for their response. In view of the responses and clarification, I am amending my score from 6 to 7.

---

> > > > ### Author Response · Authors · 2022-08-09
> > > > **Thank you**
> > > >
> > > > Thank you very much for your response and the updated score!

---

### Official Review · Reviewer_B6yz · 2022-07-09

**Rating:** 7
**Confidence:** 4
**Soundness:** 3 good
**Presentation:** 3 good
**Contribution:** 3 good

**Summary:**

The authors propose several parameterisation of the vector field in neural ODEs/CDEs/ODE-RNNs based on FWP and learning rules.

**Questions:**

I have direct questions for the authors. I have made some suggestions above; to highlight them:

1. I think the experiments could be improved by moving beyond just the CDE/RDE case.
2. The choice of numerical method and software should definitely be discussed.

**Limitations:**

There are no ethical concerns with this work.

**Strengths And Weaknesses:**

**Strength: topic**

There has been relatively little convincing work on good architectures for RNNs, NCDEs, etc. This work addresses a meaningful problem.

**Strength: originality**

To the best of my knowledge the ideas presented in this paper (combining NCDEs with FWP, etc.) are new.

The scalability, as compared to the original formulation of neural CDEs, is an important contribution.

**Strength: literature review**

This work ties together many individual lines of work -- neural CDEs, neural RDEs, long sequence modelling, learning rules, FWP, etc. -- and so far as I can determine does an excellent job providing complete references for all of them.

**Strength: experiments**

The experiments look to have been carefully conducted. They go into a great deal of detail. Moreover the results obtained demonstrate meaningful improvements on existing benchmarks.

**Strength: presentation**

The prose is carefully worded and precise. I enjoyed reading this paper.

**Weakness: experiments**

I believe the experiments are overly-focused on neural CDE and neural RDE comparisons. Whilst these are the natural choice given the framing of the paper, I think it would improve the paper to include more quantitative comparisons against other works, e.g. [0], [Ref. 34, 46, 47 from the paper]. Indeed there have been quite a few papers proposing novel architectures for RNNs/etc.

**Weakness: presentation**

At least for me -- I am largely unfamiliarS with the work on learning rules, FWP etc. -- I had to wait until the experimental section to determine that what is being proposed is essentially a new architecture, for the purposes of supervised learning etc. I would suggest explicitly stating this kind of thing up-front.

**Weakness: "the adjoint sensitivity method" (line 66)**

The term "adjoint" has become heavily overloaded. In the current literature is is often used to mean specifically optimise-then-discretise, but in older literature it was often used to mean specifically discretise-then-optimise. Following [1, Remark 5.5] I recommend avoiding it altogether.

In passing, since backpropagation is performed via OtD, then the authors may find speedups by utilising the technique of [2].

**Weakness: numerical/software**

The choice of computational framework, and choice of numerical solvers, seems to go entirely undiscussed. (I did not see it mentioned in the appendix either.)

This is a meaningful limitation: the choice of numerical solver, its choice of tolerance, etc., can greatly affect results.

**Minor issues**

Line 33: I wouldn't use the word "ancient". Some of us were around back then!

Line 78: I would argue that natural cubic splines should not be mentioned here. As [Ref. 31 of this paper] shows, there are now much better alternatives, so I think it would be better to avoid discussing natural cubic splines altogether. (e.g. they are not even implemented in [3])

**Commentary**

Line 85: I'm not sure I completely agree with this analysis. As per the next heading ("Second") then the absolute value of $x$ may be treated as an input as well, by recording and replaying it. In practice it's not been demonstrated that this actually occurs in practice, so I think it possible that this paper's side-note may be true in practice, if not in theory.

I quite like the idea of directly providing $x$ or $x'$ to the vector field, in addition to leaving $x'$ outside, as in equation (23). Most works have usually framed the inside/outside placement of $x$ or $x'$ as a dichotomy.

**References**

[0] Coupled Oscillatory Recurrent Neural  Network (coRNN): An accurate and (gradient) stable architecture for  learning long time dependencies, Rusch and Mishra, ICLR 2021

[1] On Neural Differential Equations, Kidger, Doctoral Thesis, University of Oxford 2021

[2] "Hey, that's not an ODE": Faster ODE Adjoints via Seminorms, Kidger et al, ICML 2021

[3] Diffrax, https://github.com/patrick-kidger/diffrax

---

> ### Author Response · Authors · 2022-08-01
> **Response to Reviewer B6yz**
>
> We thank the reviewer for many very positive comments.
>
> > *I believe the experiments are overly-focused on neural CDE and neural RDE comparisons. Whilst these are the natural choice given the framing of the paper, I think it would improve the paper to include more quantitative comparisons against other works, e.g. [0], [Ref. 34, 46, 47 from the paper]. Indeed there have been quite a few papers proposing novel architectures for RNNs/etc.*
>
> Yes, our main focus is clearly on the comparison/advancement of the differential equation based sequence processing models. Also, as a matter of style, we find it  much more elegant and concise to focus on the specific comparisons we want to show and avoid a huge table with all numbers mixing all comparisons. This having said, we do already compare with the strong ODE-inspired discrete-time model RNN of Ref 47 (46 and 47 are from the same authors and 47 is the follow-up of 46) in line 284 “This almost matches the state-of-the-art…”). In the revision, we’ll also add a sentence to comment on/compare with the GRU-D result as suggested by Reviewer 95a7.
> If the reviewer has specific suggestions, we'll be happy to add more numbers.
>
> > *At least for me -- I am largely unfamiliar with the work on learning rules, FWP etc. -- I had to wait until the experimental section to determine that what is being proposed is essentially a new architecture, for the purposes of supervised learning etc. I would suggest explicitly stating this kind of thing up-front.*
>
> The paragraphs in the introduction (one starting in line 38 and the following one from line 44) are essentially dedicated to this purpose (line 38: "obtain a new class of sequence processing Neural ODEs which are continuous-time counterparts of Fast Weight Programmers and linear Transformers.", line 44: "conduct experiments on three standard time series classification tasks"). We are unsure if the reviewer has somehow missed this part or if s/he still finds this not "upfront" or "explicit" enough. If it’s the latter, we’d be happy to consider further edits.
>
> > *The term "adjoint" has become heavily overloaded. In the current literature is is often used to mean specifically optimise-then-discretise, but in older literature it was often used to mean specifically discretise-then-optimise. Following [1, Remark 5.5] I recommend avoiding it altogether.*
>
> Thank you for this valuable comment. In the revision, we will replace all instances of "adjoint method" by "continuous adjoint method" to avoid confusions with older literature.
>
> > *In passing, since backpropagation is performed via OtD, then the authors may find speedups by utilising the technique of [2].*
>
> Thank you for pointing this out. We will consider this in future work.
>
> > *The choice of computational framework, and choice of numerical solvers, seems to go entirely undiscussed. (I did not see it mentioned in the appendix either.) This is a meaningful limitation: the choice of numerical solver, its choice of tolerance, etc., can greatly affect results.* // *2. The choice of numerical method and software should definitely be discussed.*
>
> Thank you for pointing this out. Essentially we used the same configurations used in the baseline models from the prior works. We will add the corresponding details (which were only implicitly provided via our code) in the appendix in the revision. We also plan to thank the authors of the base code ours is based on in the acknowledgement in the final version if the paper is accepted.
>
> > *Line 33: I wouldn't use the word "ancient". Some of us were around back then!*
>
> We apologise for this. Reviewer 95a7 made a similar comment. We did not realise that the word "ancient" had such a negative connotation. We will simply remove it.
>
> > *Line 78: I would argue that natural cubic splines should not be mentioned here. As [Ref. 31 of this paper] shows, there are now much better alternatives, so I think it would be better to avoid discussing natural cubic splines altogether. (e.g. they are not even implemented in [3])*
>
> Thank you for pointing this out. We will remove the corresponding mention.
>
> > *Line 85: I'm not sure I completely agree with this analysis. As per the next heading ("Second") then the absolute value of x may be treated as an input as well, by recording and replaying it. In practice it's not been demonstrated that this actually occurs in practice, so I think it possible that this paper's side-note may be true in practice, if not in theory.*
>
> Since this seems to be just a comment in passing, we will leave it as is (please correct us if we misunderstood your point).
>
> We hope that our response successfully addresses the reviewer's comments.
> If that is the case, and if you think this paper should be accepted, we'll appreciate it a lot if you can consider increasing the score. Thank you.

---

> > ### Comment · Reviewer_B6yz · 2022-08-07
> > **Response**
> >
> > I am pleased to say that I find the authors' response quite convincing, and have updated my score 6->7 in response.
> >
> > Regarding clarity on learning rules etc. -- I do not think the lines highlighted are sufficiently clear. I really am looking for something explicit, to the effect of "we propose a new architecture, for the purposes of supervised learning".

---

> > > ### Author Response · Authors · 2022-08-07
> > > **Thank you**
> > >
> > > Thank you very much for your response and the updated score.
> > >
> > > We will make the corresponding text more explicit as requested.

---

### Official Review · Reviewer_mFC6 · 2022-07-13

**Rating:** 5
**Confidence:** 4
**Soundness:** 2 fair
**Presentation:** 3 good
**Contribution:** 3 good

**Summary:**

The paper presents a continuous time version for Fast Weight Programmers family of sequence classification models. It presents continuous time analogues to Hebbian learning rules. It introduces continuous time versions of multiple FWPs variants. The continuous time versions are empirically tested three standard sequence classification tasks, and compared to other sequence classifiers. In the appendix, the method is also applied to two basic RL problems.


**Questions:**

### Experiments:

In the experiments, why were these particular baselines chosen? Why were only ODE based methods chosen as comparison? Could a non-ode method be included in these problems?

Tables 1 & 2:  Are the models comparable? What are the parameter counts of each row? What are the runtimes of training and inference? What are the architecture hyperparameters?

Table 2: Are the results for “Direct NODE” or “NCDE”?

Line 264: Did the authors perform the experiment comparing NODE and NCDE equations for this problem? It is not clear from the text or results.

### Usage of “memory efficient” adjoint

As mentioned, it is not clear that the checkpoint-free adjoint method can be used to these equations. Are the equations of the continuous FWP models reversible, and guaranteed to remain so?

Is the adjoint method only applied on the fast weight learning rule, or the entire network?

Did you compare using the checkpoint-free adjoint method vs. regular backpropagation in your examples?

Were the baseline methods also able to use the “memory saving” adjoint method for training?

### Clarifications

Line 211: Why is it not practical? Is it because of the number of parameters, or computational cost, something else?

Line 217: Is the Latent ODE RFWP used anywhere in this work?
Line 218: This last sentence is confusing. “While this case” Which is “this case”? Does this paragraph mean to say that all of section 3.3 is not of central interest?

Line 263: What is an RDE?

### Minor

This is a very superficial comment, but the paper title makes it difficult to understand what the paper is about. The phrase “Learning to Program” suggests a different field of program synthesis or general optimization algorithms, which does not fit. The title could be simplified.


**Limitations:**

The authors focus on how their method mitigates limitations of the original family of methods. The authors have some discussions about the cost limitations of the family methods, and hypotheses on the mechanism by which their method performs well with low-rank methods when previous variants do not.

There are no potential negative societal impacts that are particular to their work.


**Strengths And Weaknesses:**

The paper is overall well presented with a thorough explanation. The continuous analogous of FWPs is an original contribution to the literature of neural ODEs. A particular novelty is the consideration of Hebbian learning methods.

A wide variety of experiments are performed, which is a strength of the paper. However, the main weakness of this paper is that each experiment only has one baseline method as comparison, and does not have enough details, which makes it unclear if the proposed method is significantly better than existing methods.

This is a minor aspect of the work: the “memory saving” adjoint method is mentioned as a “pro” of their method. A questionable claim is “Memory Efficient Backpropagation for FWPs”. The “memory-saving” adjoint method only works if the equation is reversible (not all ODEs can be integrated in both directions c.f. heat equation). Otherwise, checkpointing of the forward pass of the ODE is required.   (See, e.g. Zhuang et al. “Adaptive Checkpoint Adjoint Method for Gradient Estimation in Neural ODE'' 2020). Additionally, the baseline methods could also use it (with the same error) so it is not a pro unique to their method.

---

> ### Author Response · Authors · 2022-08-01
> **Response to Reviewer mFC6, part 1/3**
>
> We thank the reviewer for positive comments on the strengths of this work ("original contribution", "thorough explanation" & "a wide variety of experiments").
> Regarding the concerns, we believe we have good explanations to resolve all of them.
> If you find our response convincing, we’d appreciate it a lot if you could consider increasing the score. Please find our response inline (we grouped redundant/similar questions from different sections.).
>
> > *However, the main weakness of this paper is that each experiment only has one baseline method as comparison, and does not have enough details, which makes it unclear if the proposed method is significantly better than existing methods.* // *In the experiments, why were these particular baselines chosen? Why were only ODE based methods chosen as comparison? Could a non-ode method be included in these problems?*
>
> One of the main contributions of this work is to advance the rather recent research trend of continuous-time sequence processing models based on neural differential equations, by proposing new models belonging to this family which are the continuous counterparts of linear Transformers/Fast Weight Programmers with a better scalability property (Sec 5 from line 291). Since the goal of our experiments is to show that our models also empirically outperform the best existing NCDE model, our baseline is naturally the standard NCDE (including its descendent NRDE for very long sequence processing; Table 2) which is currently the best approach for DE-based continuous-time sequence processing (as continuous counterparts of RNNs). The comparison to these baselines is sufficient to conclude that our models yield the best performance among DE-based sequence processing models.
>
> Please note that, where relevant (see line 284, “This almost matches the state-of-the-art…”), we do also compare with a state-of-the-art discrete-time model. Following the remark of Reviewer 95a7, we will also add a comparison with the GRU-D in the PhysioNet Sepsis experiments in the revision.
>
> Regarding *“does not have enough details”*, please note that all important experimental details (e.g., hyper-parameters) are provided in Appendix B. In addition, the supplemental material contains all code needed to reproduce our results (which we will make public if the paper is accepted). We really care about  transparency, and this has been noted by Reviewer 95a7 as one of the strengths of this work.
>
> > *This is a minor aspect of the work: the “memory saving” adjoint method is mentioned as a “pro” of their method. A questionable claim is “Memory Efficient Backpropagation for FWPs”. The “memory-saving” adjoint method only works if the equation is reversible (not all ODEs can be integrated in both directions c.f. heat equation). Otherwise, checkpointing of the forward pass of the ODE is required. (See, e.g. Zhuang et al. “Adaptive Checkpoint Adjoint Method for Gradient Estimation in Neural ODE'' 2020). Additionally, the baseline methods could also use it (with the same error) so it is not a pro unique to their method.* // *As mentioned, it is not clear that the checkpoint-free adjoint method can be used to these equations. Are the equations of the continuous FWP models reversible, and guaranteed to remain so?*
>
> First of all, there is a misunderstanding about our claim. We never state that the memory efficiency is a unique "pro" of our method (if you find any confusing sentences, please let us know the line number explicitly). We clearly state in the background section (see lines 80-82) that all ODE based models (not only ours) benefit from this continuous adjoint method. What we stress instead (in Sec 5) is that this memory efficiency which is "optional" for the baseline NCDE is crucial for our fast weight programmers, since naive backpropagation is impracticable for our models, as is also the case in the discrete case. And yes, the use of the continuous adjoint method for our models is mathematically valid. As we state in Sec 3, all our models remain within the framework of Neural ODEs/CDEs (see the canonical form in Eqs 18 \& 23). The only things we change are the model architecture/parameterisation of the neural vector field.
>
> In passing, we note that the heat equation is not an ODE. Also, the reviewer’s statement *“otherwise, checkpointing of the forward pass of the ODE is required”* is not correct. While checkpointing may help reducing the error of backward pass, the whole concept of “Adaptive Checkpoint Adjoint method” proposed by Zhuang et al. is still based on the continuous adjoint method (as its name indicates); it is a helper to the adjoint method, not an alternative. If the continuous adjoint method was not valid, checkpointing would also not make any sense.
> Overall, can it be the case that the reviewer was conceptually confused by these technical subtleties at the time of writing?

---

> > ### Author Response · Authors · 2022-08-01
> > **Response to Reviewer mFC6, part 2/3**
> >
> > > *Is the adjoint method only applied on the fast weight learning rule, or the entire network?*
> >
> > Please note that the continuous adjoint method is a method to computing the backward pass of the ODE layers. Thus, we only use it for the fast weight learning rule (for the rest of the model, the standard backpropagation is used to compute all gradients needed).
> >
> > > *Did you compare using the checkpoint-free adjoint method vs. regular backpropagation in your examples?*
> >
> > No, as we stress in Sec 5, just like with the discrete FWPs, naive implementation (e.g. PyTorch automatic differentiation) of the regular backpropagation would result in prohibitive memory requirement. In the discrete case, this is circumvented by a custom backward implementation (in CUDA), which is not easy in combination with ODEs.
> > As shown experimentally, our models yield good performance using the standard continuous adjoint method.
> >
> > > *Were the baseline methods also able to use the “memory saving” adjoint method for training?*
> >
> > Yes, as we note above and explain in the background section, the baselines also make use of the memory efficient continuous adjoint method.
> >
> > > *Tables 1 \& 2: Are the models comparable? What are the parameter counts of each row? What are the runtimes of training and inference? What are the architecture hyperparameters?*
> >
> > Similar to the baseline models, we conduct hyper-parameter search in the ranges we specify in Appendix B. We even use the same codebase as the baselines which ensures that the evaluation metrics (as well as train/test splits) are comparable. In this sense, yes, they are all comparable.
> >
> > Please note that we also already report parameter counts and inference time where it is most relevant (see line 287 “The wall clock time…”). If the reviewer wishes, we can provide the numbers for all models in the revision, but please note that comparing the exact parameter count or inference time has limited meaning here, because we select the best model simply by looking at its validation performance. That means that if Configuration A has a parameter count twice larger than Configuration B, even if A achieves only 0.1 point better performance than B, we would still select A as our best model. We did not optimise for other metrics than the validation performance since our goal is to simply show that our proposed models outperform the best existing NCDE models.
> >
> > > *Table 2: Are the results for “Direct NODE” or “NCDE”?
> > Line 264: Did the authors perform the experiment comparing NODE and NCDE equations for this problem? It is not clear from the text or results.*
> >
> > For consistency with the prior setting of Morrill et al [32], in the experiments for Table 2, the input to the network is either log-signature (top part of Table 2) or x’ (bottom part of Table 2). In these cases, direct NODE and NCDE/RDE coincide. This is actually what we note in footnote 4 ("As a sidenote...") and line 263 ("The equations..."), and we also remind the reader of the relation between NCDE and NRDE in line 270 (“When set to 1, …”).
> > However, we agree with the reviewer that this aspect is a bit subtle: we will improve the clarity in the revision.
> >
> > > *Line 211: Why is it not practical? Is it because of the number of parameters, or computational cost, something else?*
> >
> > Parameterising a high-dimensional weight matrix as output of an NN is impractical in terms of both computational costs and number of parameters. We did not explain this further as this is exactly the same scalability bottleneck as the baseline NCDE, but since the reviewer finds this confusing, we will add further clarifications in the revision. Thank you for pointing this out.
> >
> > > *Line 217: Is the Latent ODE RFWP used anywhere in this work? Line 218: This last sentence is confusing. “While this case” Which is “this case”? Does this paragraph mean to say that all of section 3.3 is not of central interest?*
> >
> > We introduce a case distinction in the background section (line 72 "First," line 91 "Second," line 98 "Finally"), and our Sec 3 follows this structure. The "case" in "while this case" corresponds to this third case.
> > Yes, the entire Sec 3.3 is not of "central interest" (in the sense that the main contributions of this paper would remain without this Sec. 3.3). However, it would be unnatural to ignore this third case. We therefore decided to conceptually cover this case in the main text for the sake of completeness, and to dedicate an experimental section in Appendix C.3 where the Latent ODE RFWP is tested in the reinforcement learning setting (which falls into this third case). Actually, at the beginning the reviewer wrote: “In the appendix, the method is also applied to two basic RL problems”, so it seems that the reviewer already did note this (but maybe after s/he wrote this question?).

---

> > > ### Author Response · Authors · 2022-08-01
> > > **Response to Reviewer mFC6, part 3/3**
> > >
> > > > *Line 263: What is an RDE?*
> > >
> > > RDE stands for Rough Differential Equation (as we introduce in line 84 in the background section; same as in the original paper by Morrill et al. [32]) which is one of our baseline models in Table 2.
> > >
> > > > *Minor. This is a very superficial comment, but the paper title makes it difficult to understand what the paper is about. The phrase “Learning to Program” suggests a different field of program synthesis or general optimization algorithms, which does not fit. The title could be simplified.*
> > >
> > > The exact phrase is "Learning to Program Neural Nets" which refers to the previous line of work on fast weight programmers (an end-to-end differentiable neural network that learns to program other nets by modifying their weight matrices).
> > >
> > > We hope that our response resolves the reviewer’s main concerns. If that is the case, please consider increasing the score. If not, we really would like the reviewer to clarify the reason to rank soundness as "2" when our work is a clear advancement (empirical performance and scalability) in the popular NCDE framework.

---

> > > > ### Author Response · Authors · 2022-08-09
> > > > **Friendly reminder**
> > > >
> > > > This is just a friendly reminder about the NeurIPS rebuttal deadline today. Thank you!

---

### Author Response · Authors · 2022-08-01
**General response**

We thank all the reviewers for their valuable comments.
We reply to each reviewer individually below.
Please note that we will apply all edits we promise in this rebuttal if the paper is accepted.
For the moment, we opt for keeping the submitted PDF as is, in order to avoid changing the original line numbering, and such that we can refer to originally submitted texts.
Overall, we believe that our response should resolve all main concerns raised by the reviewers.
If you find our explanations convincing, please consider increasing your scores. Thank you.

---

### Meta-Review · Area_Chair_5Fcx · 2022-08-26

**Recommendation:** Accept
**Confidence:** Certain

**Metareview:**

A novel blend of linear transformers and "Fast Weight Programmers" is proposed where slow weights generate an embedding and learning rule parameterizations of a Neural ODE based evolution of "fast weights" of another network. The new architectures appear to provide superior performance on several benchmarks in comparison to NCDE/NRDE benchmarks. The reviewers agree on the core contributions and recommend strengthening some presentation aspects of the paper, possibly reorganizing the main and supplementary materials so the paper is more self-contained.

**Award:**

No

---

### Decision · Program_Chairs · 2022-09-14

Accept